

# Finite-temperature photoemission in the extended Falicov-Kimball model: a case study for Ta$_2$NiSe$_5$

## Satoshi Ejima$^\star$, Florian Lange and Holger Fehske

Institute of Physics, University Greifswald, 17489 Greifswald, Germany

$\star$ ejima@physik.uni-greifswald.de

## Abstract

Utilizing the unbiased time-dependent density-matrix renormalization group technique, we examine the photoemission spectra in the extended Falicov-Kimball model at zero and finite temperatures, particularly with regard to the excitonic insulator state most likely observed in the quasi-one-dimensional material Ta$_2$NiSe$_5$. Working with infinite boundary conditions, we are able to simulate all dynamical correlation functions directly in the thermodynamic limit. For model parameters best suited for Ta$_2$NiSe$_5$ the photoemission spectra show a weak but clearly visible two-peak structure, around the Fermi momenta $k \simeq \pm k_{\mathrm{F}}$, which suggests that Ta$_2$NiSe$_5$ develops an excitonic insulator of BCS-like type. At higher temperatures, the leakage of the conduction-electron band beyond the Fermi energy becomes distinct, which provides a possible explanation for the bare non-interacting band structure seen in time- and angle-resolved photoemission spectroscopy experiments.


---

# 1   Introduction

In solid state physics, superconductivity is a classic example of emerging quantum coherence on a macroscopic scale. An essential prerequisite for this is the pairing of electrons, normally of opposite spin and close to Fermi surface of metals. Similarly, valence-band holes and conducting-band electrons may form excitonic pairs in semimetals with small band overlap or in semiconductors with a small band gap, triggered by their Coulomb attraction, where the excitons are expected to "condense" in a macroscopic coherent state at low temperatures under restrictive conditions [1–5]. This state is called excitonic insulator (EI) because it exhibits no "super-transport" properties. It is stabilized by the opening of a gap in semimetals or by the flattening of the valance-band top (conduction-band minimum) in semiconductors, which both can be detected by angle-resolved photoemission spectroscopy (ARPES). Usually exciton condensation in a semiconductor setting is discussed in terms of Bose-Einstein condensation (BEC), while those appearing for a semimetallic (noninteracting) band structure should be described in close analogy with the Bardeen-Cooper-Schrieffer (BCS) theory [4–9].

Even though the obvious EI scenario was predicted almost 60 years ago, it was only recently that some material classes have shown some signatures of an EI ground state. In this respect one of the most promising candidates seems to be the quasi-one-dimensional material $Ta_2NiSe_5$, which possesses a tiny direct gap at the $\Gamma$ point of the Brillouin zone. At a critical temperature $T_c \approx 328\,K$, $Ta_2NiSe_5$ encounters a second-order transition from an orthorhombic ($T > T_c$) to a monoclinic ($T < T_c$) structure [10]. ARPES experiments revealed a flattening of the valence band when cooling below the transition temperature [11, 12], which has been taken as indication of an EI state within a model simulation [13]. Motivated by this, various experiments have recently performed for this material, using NMR [14], inelastic x-ray scattering [15], scanning tunneling microscopy or spectroscopy [16, 17], and time- and angle-resolved photoemission spectroscopy (t-ARPES) [18–20]. Besides there is the isostructual compound $Ta_2NiS_5$, which opens up the possibility to perform comparative measurements and analysis, not only of the photoemission [21] but also of the magnetic susceptibility [10] and the optical conductivity [22, 23].

Unfortunately, some experimental findings and theoretical predictions are inconsistent so far. For instance, the optical conductivity spectra in $Ta_2NiSe_5$ [23] show an extra peak in the low-energy regime (at about $\omega \simeq 0.4$ eV) that cannot be reproduced by the density-functional-theory-based simulation [24]. The authors of Ref. [23] attributed this low-energy peak to the giant oscillator strength of spatially extended exciton-phonon bound states, while the theoretical work [24] concluded that the interorbital Coulomb interaction between valence and conduction bands should be sufficient to explain this peak. Another important issue is whether the formation of the EI in $Ta_2NiSe_5$ follows a BCS or a BEC scenario. While the small value of the transport gap [22] implies that the excitonic state in $Ta_2NiSe_5$ is of BCS type, the spectral weight broadening of the flat band with increasing temperature has been taken as a signature of a breaking of the quasiparticle peak structure, i.e., of dissolving a Bose-Einstein condensate [13].

In contrast to the advanced experimental investigations performed for $Ta_2NiSe_5$ in the last

few years, most of the theoretical studies are mainly based on weak-coupling approaches, especially if the single-particle spectra have been addressed [13, 25, 26]. When focusing on strictly one-dimensional (1D) systems this is not necessary, however, because we can calculate both ground-state and dynamical quantities for the considered model Hamiltonians by rather unbiased numerical techniques like the density-matrix renormalization group method (DMRG) [27] and its further developments for dynamical quantities [28, 29]. For sure, such a de-facto approximation-free treatment is essential to resolve some of the issues carved out above.

To this end, in this work, we re-examine the properties of the ground state of the 1D extended Falicov-Kimball model (EFKM), as well as the behavior of various spectral functions in the EFKM at zero and finite temperatures, by means of the time-dependent DMRG (t-DMRG) technique in the matrix-product-state (MPS) representation [30–33]. The EFKM at half band filling can be considered as the perhaps minimal model for Ta$_2$NiSe$_5$. We demonstrate that the best-suited parameter set typifies Ta$_2$NiSe$_5$ as an EI in the BCS regime; here, the zero-temperature photoemission spectra exhibit two-peak structure. At finite temperatures the gap melting is clearly visible in the photoemission spectra, and the leakage of the conduction band beyond the Fermi energy becomes distinct as temperature increases, which might be related to the bare band structure obtained in the t-ARPES experiment [19].

The paper is organized as follows. In section 2, we introduce the EFKM Hamiltonian, present its ground-state phase diagram, including the BCS-EI⇌BEC-EI crossover regime deduced from pair-condensation amplitude, and comment on the best model parameters used for describing Ta$_2$NiSe$_5$. Section 3 contains our numerical data for the photoemission spectra in the EFKM, which are discussed with regard to the experimental results for Ta$_2$NiSe$_5$. Our main conclusions are presented in section 3. Details of the used numerical scheme can be found in Appendix A. For comparison, Appendix B provides results for the photoemission spectra in the 1D half-filled Hubbard model.

## 2 Theoretical modeling

### 2.1 Extended Falicov-Kimball model

Let us first introduce the 1D EFKM [34–37], which has been argued to be the minimal theoretical model for Ta$_2$NiSe$_5$ [26, 38, 39]. The Hamiltonian reads

$$\hat{H} = - \sum_{\alpha=c,f} t_\alpha \sum_j \left( \hat{\alpha}_j^\dagger \hat{\alpha}_{j+1} + \text{H.c.} \right) + U \sum_j \hat{n}_j^c \hat{n}_j^f + \frac{D}{2} \sum_j \left( \hat{n}_j^c - \hat{n}_j^f \right) - \mu \sum_{\alpha,j} \hat{n}_j^\alpha, \qquad (1)$$

where $\hat{\alpha}_j^\dagger$ ($\hat{\alpha}_j$) denotes the creation (annihilation) operator of a spinless fermion in the $\alpha = \{c, f\}$ orbital at Wannier site $j$, $\hat{n}_j^\alpha = \hat{\alpha}_j^\dagger \hat{\alpha}_j$, $U$ is the local Coulomb repulsion between $c$ and $f$ electrons staying at the same lattice site, $D$ parametrizes the level splitting of $c$ and $f$ orbitals, and $\mu$ is the chemical potential. Representing the orbital flavour of the EFKM by a pseudospin variable [40, 41], $t_c \to t_\uparrow$, $t_f \to t_\downarrow$, $\hat{c}_j \to \hat{c}_{j,\uparrow}$, and $\hat{f}_j \to \hat{c}_{j,\downarrow}$, the model (1) can be viewed as asymmetric Hubbard model with spin-dependent hopping in a magnetic field. In other words, the standard Hubbard model (see. Eq. (8) in Appendix B.1) is obtained by setting $t_c = t_f$ and $D = 0$ in the EFKM. In what follows, we take $t_c$ as the unit of energy and, with a view to Ta$_2$NiSe$_5$, consider the half-filled band case only. Note that direct $f$-$c$ hopping is prohibited by symmetry for this material [26].

The DMRG ground-state phase diagram of the half-filled 1D EFKM has been worked out previously [37]. Depending on the strength of the orbital level splitting, the system realizes one of three insulating phases: a state characterized by staggered orbital order (SOO) which

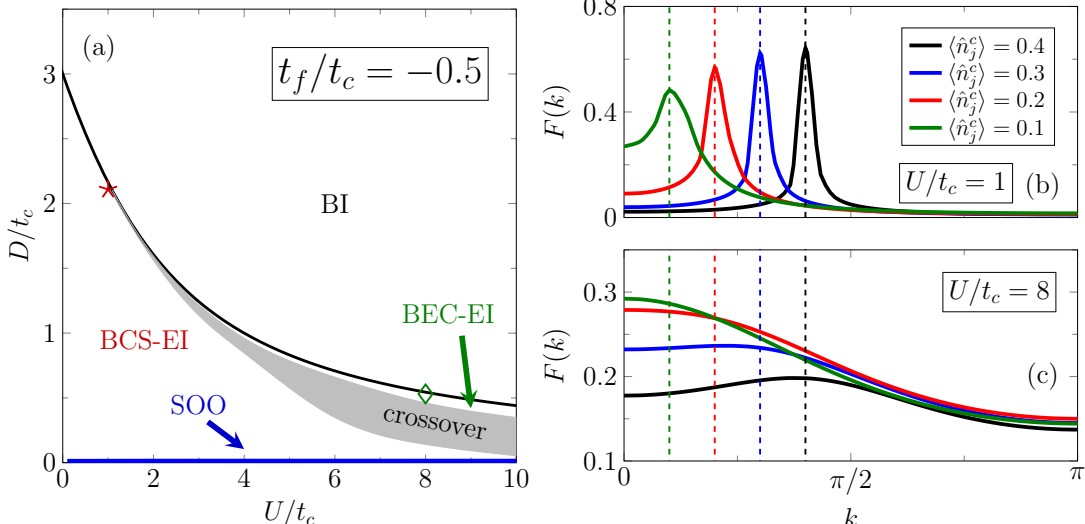

Figure 1: (a) DMRG ground-state phase diagram of the 1D half-filled EFKM, showing an excitonic insulator (EI) state (of BCS- respectively BEC-type) sandwiched between band insulator (BI) and staggered orbital order (SOO) phases. Star ($\star$) and diamond ($\diamond$) symbols mark the parameter sets that will be used in simulations of the spectral functions $A(k, \omega)$ in Sec. 3. (b) and (c): Condensation amplitudes $F(k)$ at various $\langle \hat{n}_j^c \rangle$ for $U/t_c = 1$ and 8. Data obtained by DMRG with periodic boundary conditions for $L = 60$ (we checked that for such a large system finite-size effects are negligible). Dashed lines give the corresponding Fermi momenta $k_F = \pi \langle \hat{n}_j^c \rangle$ in the noninteracting limit. In all cases, $t_f/t_c = -0.5$.

corresponds to an Ising-like antiferromagnet in the strong-coupling limit [34,42], the EI where the $c$-$f$ electron coherence emerges (here, $0 < \langle \hat{n}_j^c \rangle < 1/2 < \langle \hat{n}_j^f \rangle < 1$), and the (rather trivial) band insulator (BI) with $\langle \hat{n}_j^c \rangle = 0$ and $\langle \hat{n}_j^f \rangle = 1$. While the EI-BI phase boundary is given analytically [43],

$$D_{c_2} = \sqrt{4\left(|t_f| + |t_c|\right)^2 + U^2} - U,\tag{2}$$

the SOO-EI quantum phase transition has to be determined numerically, e.g., from

$$D_{c_1} = E_0(L/2 + 1, L/2 - 1) - E_0(L/2, L/2),\tag{3}$$

where $E_0(N_f, N_c)$ is the ground-state energy for a finite system with $L$ lattice sites, $N_f$ $f$-electrons, and $N_c$ $c$-electrons.

Figure 1(a) shows the ground-state phase diagram of the 1D EFKM in the $U$-$D$ plane for $t_f/t_c = -0.5$. Obviously, the SOO phase takes up only a small space in the vicinity of $D = 0$. Its boundary (3) can be easily extracted from DMRG calculation of the various ground-state energies at fixed system sizes, supplemented by a finite-size extrapolation to the thermodynamic limit [37]. The EI and BI phases are separated by the transition line (2). We note that mean-field [44, 45] and slave-boson [35, 36] approaches cannot be used to determine the EI phase in the 1D EFKM system because there is no continuous symmetry that is broken. As a result, we do not have a suitable order parameter. For example, the expectation value $\langle \hat{c}^\dagger f \rangle$, which serves as excitonic order parameter in higher dimensions, becomes zero in the limit of vanishing (explicit) $c$-$f$-band hybridization [46]. However, critical excitonic correlations can be detected in a certain parameter regime of the 1D EFKM [34]. Exploiting the off-diagonal

anomalous Green function scheme [47], these correlations are captured by the condensation amplitude

$$F(k) = \langle \psi_1 | \hat{c}_k^\dagger \hat{f}_k | \psi_0 \rangle \tag{4}$$

in the momentum space. In (4), $|\psi_0\rangle$ is the ground state for a finite system with $L$ lattice sites and $N_f$ ($N_c$) $f$-electrons ($c$-electrons); $|\psi_1\rangle$ is the excited state with ($N_f - 1$) $f$-electrons and ($N_c + 1$) $c$-electrons. If the electron-hole pairs are only weakly bound ($E_B/t_c \ll 1$, BCS-EI regime), $F(k)$ develops a sharp peak at the Fermi momentum $k_F = \pi \langle \hat{n}_j^c \rangle$ [cf. Fig. 1(b) for $U/t_c = 1$]. Here, "Fermi surface" effects play an important role. On the other hand, for tightly bound excitons ($E_B/t_c \gg 1$, BEC-EI regime), $F(k)$ has a (broad) maximum at $k = 0$, see $F(k)$ for $\langle \hat{n}_j^c \rangle = 0.1$ and $0.2$ at $U/t_c = 8$ in Fig. 1(c). One can therefore define a BCS-BEC crossover region, where $F(k)$ shows a maximum for $0 < k < k_F$. Within the EI, the respective defined BCS-BEC crossover range is visualized in Fig. 1(a) by the shaded region. We see that for the $t_f/t_c = -0.5$ ratio used, an EI of BEC type appears for $U/t_c \gtrsim 3$ in the vicinity of the EI-BI transition line only. In this respect we note that for $U/t_c = 1$ [$U/t_c = 8$] the EI-BI transition occurs at $D_{c_2}/t_c \simeq 2.16$ [$D_{c_2}/t_c \simeq 0.54$], and the results depicted in Fig. 1(b) [Fig. 1(c)] for $\langle \hat{n}_j^c \rangle = 0.1$, $0.2$, $0.3$, and $0.4$ correspond to $D/t_c \simeq 2.10$, $1.85$, $1.37$, and $0.73$ [$D/t_c \simeq 0.53$, $0.47$, $0.37$, and $0.21$], respectively. The nature of the EI is further characterized by a fast, almost exponential (slow power-law) decay of the exciton-exciton correlations and a finite (divergent) exitonic momentum distribution function in the weak-coupling BCS (strong-coupling BEC) regime, see Ref. [37].

## 2.2 Ta$_2$NiSe$_5$ model parameters

To determine the optimal parameter set for an (extended) Falicov-Kimball-model-based description of Ta$_2$NiSe$_5$, ARPES data is mainly used. For this, Seki *et al.* [13] utilized a temperature-dependent variational cluster approach for the three-dimensional EFKM, assuming $t_c = -t_f$, however, not least because of the reduced computational costs due to particle-hole symmetry in this case. A perhaps more realistic approach was put forward by Kaneko *et al.* [26] who considered a three-chain electron-phonon-coupled system and exploited band-structure calculations together with a mean-field analysis. Here, the estimated parameter set is $t_c = 0.8$, $t_f = -0.4$, $U \simeq 0.55$ and $D = 0.2$ (in units of eV). In all these efforts, it is worth observing that band flattening detected in the ARPES experiments only occurs in a relative narrow region of momentum space, specifically for $|k_x| \lesssim 0.1\text{Å}^{-1}$ (taking into account that the lattice constant of the chain direction is $a = 3.51232$ Å [48], this means for $|k|/a \lesssim 0.35$). Since for $t_f/t_c = -0.5$ and $U/t_c \lesssim 1$ the system is in the EI-BCS regime where the maxima of $F(k)$ are almost equal to the peak position of the single-particle spectral functions [37], the flat band can appear only for $|k| \lesssim k_F$. This also implies that the strength of the level splitting should be very close to the EI-BI phase transition line, as for $\langle \hat{n}_j^c \rangle = 0.1$ where $k_F = 0.1\pi(< 0.35)$. Therefore, in this study, we discuss the spectral functions of the 1D EFKM using the parameter set $t_f/t_c = -0.5$, $U/t_c = 1$ (which is slightly larger than above estimation $U/t_c = 0.55/0.8 \simeq 0.69$), and $D/t_c = 2.11$. This allows us to carry out the DMRG simulations for a mean $c$-electron density $\langle \hat{n}_j^c \rangle = 1/10$ at $T = 0$. This parameter set is marked by a star ($\star$) in Fig. 1(a).

To provide a contrasting perspective, we will also consider the strong-coupling regime, specifically for $U/t_c = 8$, where the BEC-type EI is realized in a wider region of the density (i.e., for $\langle \hat{n}_j^c \rangle \lesssim 0.25$). Here, we use $D/t_c = 0.53$, and again $\langle \hat{n}_j^c \rangle = 1/10$, for the calculation of the spectral functions at $T = 0$ [cf. the $\diamond$ symbol in Fig. 1(a)].

Let us finally emphasize that we will approximate the doubly-degenerate conduction $c$-electron bands [26] by a single band and neglect any electron-phonon coupling effects [49] on the EI formation.

# 3 Numerical results for the extended Falicov-Kimball model

## 3.1 Spectral functions

The full single-particle spectrum $A(k, \omega)$ consists of two parts, $A(k, \omega) = A^-(k, \omega) + A^+(k, \omega)$, where $A^-(k, \omega)$ and $A^+(k, \omega)$ denote the photoemission spectrum (PES) and inverse PES (IPES), respectively. In the case of the EFKM, $A^\pm(k, \omega)$ splits into contributions from the $c$ and $f$ electrons, i.e., $A^\pm(k, \omega) = \sum_{\alpha=c,f} A^\pm_\alpha(k, \omega)$. As a result, to determine $A(k, \omega)$ for $t_c \neq |t_f|$, we have to compute four dynamical correlation functions, $A^\pm_\alpha(x, t)$ with $\alpha \in \{c, f\}$. These circumstances distinguish the EFKM from the simple Hubbard model, where only one of the four correlation functions $A^\pm_\sigma(x, t)$ with $\sigma \in \{\uparrow, \downarrow\}$ is needed to obtain the full spectra at half filling owing to the particle-hole and spin-flip symmetries. To obtain $A^\pm(k, \omega)$ numerically, we simulate the dynamic correlation function in real space

$$A^\pm(x, t) = \langle e^{i\hat{H}t} \hat{O}^\dagger_{j+x} e^{-i\hat{H}t} \hat{O}_j \rangle_\beta, \tag{5}$$

and then carry out a Fourier transform

$$A^\pm(k, \omega) = \sum_x e^{-ikx} \int_{-\infty}^{\infty} dt\, e^{i\omega t} e^{-\eta_L |t|} A^\pm(x, t), \tag{6}$$

where $\langle \cdot \rangle_\beta$ denotes the expectation value at inverse temperature $\beta$. In doing so, we exploit the t-DMRG technique together with the purification method [50, 51] for MPS and so-called infinite boundary conditions (IBC) [52]. The damping factor $e^{-\eta_L |t|}$ needed in Eq. (6) because of the finite simulation time leads to a Lorentzian broadening in the frequency space. Moreover, within t-DMRG framework, we use a second-order Trotter-Suzuki decomposition with a time step $\tau = \tau_\beta = 0.01$ for the real- and imaginary-time evolutions and a maximum bond dimension of 800, so that the truncation error will be smaller than $10^{-6}$ in all the calculations presented. Appendix A gives a more detailed description of our numerical scheme, whose accuracy is tested for the Hubbard model in Appendix B. .

## 3.2 Single-particle spectra at zero temperature

We now discuss the single-particle spectrum of the EFKM at half-filling for both zero ($T = 0$) and finite ($T = 1/\beta$) temperatures with a particular focus on the flattening of the band structure observed in ARPES experiments for the quasi-1D potential EI material $Ta_2NiSe_5$ around the $\Gamma$ point. Although in our strictly 1D model system the excitonic correlations will become critical at $T = 0$ only, the spectral properties of the 1D EFKM at low but finite temperatures will reflect the existence and nature of the zero-temperature EI state.

Figure 2 displays the zero-temperature, wave-vector- and energy-resolved spectral functions in the EFKM, using the parameter set best-suited for $Ta_2NiSe_5$: $t_f/t_c = -0.5$, $U/t_c = 1$ and $D/t_c = 2.11$ [marked by the star in Fig. 1(a)]. Here, panels (a) and (b) give the $c$- and $f$-orbital parts of the spectral functions $A_c(k, \omega) = A^+_c(k, \omega) + A^-_c(k, \omega)$ and $A_f(k, \omega) = A^+_f(k, \omega) + A^-_f(k, \omega)$, respectively, which are combined in panel (c). In this regime, starting out from a semi-metallic bare band situation, the EFKM realizes an EI phase of BCS type built up from weakly-bound electron-hole pairs. Since the Coulomb attraction is relatively weak, $A_c(k, \omega)$ and $A_f(k, \omega)$ almost follow the unrenormalized $c$- and $f$-band dispersions. Concomitantly, we find a more or less uniform distribution of the spectral weight and weak incoherent contributions close to band edge only. Nevertheless, a gap develops at the "bare" Fermi momentum $k_F = \pi \langle \hat{n}^c_j \rangle$, which is exponentially small, however, and therefore difficult to detect in panels (a)-(c) in view of the used Lorentzian broadening $\eta_L/t_c = 0.1$. It is actually only confirmed

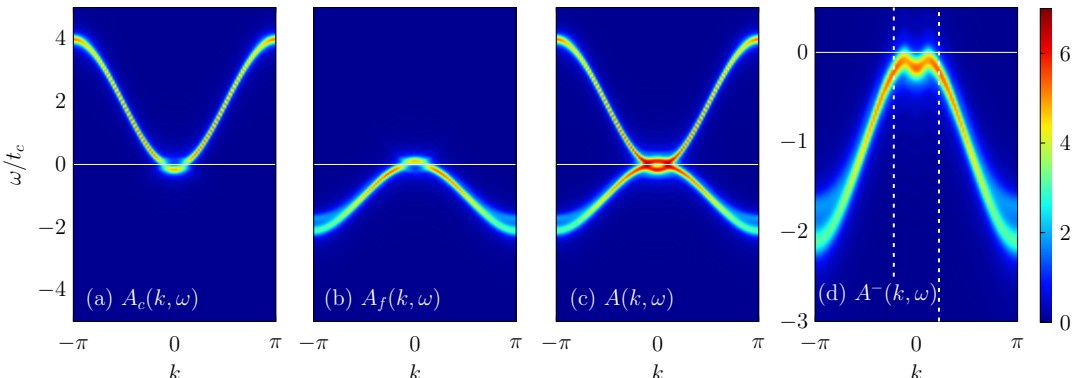

Figure 2: Zero-temperature single-particle spectral functions $A_\alpha(k,\omega)$, $A(k,\omega)$ and $A^-(k,\omega)$ in the 1D half-filled EFKM with $t_f/t_c = -0.5$, $D/t_c = 2.11$, and $U/t_c = 1$. The mean $c$-electron density is fixed to be $\langle \hat{n}_j^c \rangle = 0.1$. The dashed lines in panel (d) mark the momentum interval considered in ARPES experiments on $Ta_2NiSe_5$ [11]. Results are obtained by t-DMRG using IBC and a window size $L_W = 200$ to keep the values of the dynamical correlation functions on the boundaries less than $10^{-8}$ up to the target time $t_{fin} \cdot t_c = 16$. We note that the unit cell for the iDMRG simulations needs to be large ($N_{uc} = 10$) to realize the small $c$-electron density $\langle \hat{n}_j^c \rangle = 0.1$.

in the zoomed-in panel for the photoemission part [Fig. 2(d)]. Most notably the PES shows an "M"-shaped band structure with two peaks close to $\pm k_F$, just as detected in the $Ta_2NiSe_5$–ARPES experiments at low temperatures in the vicinity of the $\Gamma$ point [13, 20], even though the energy minimum at $k = 0$ turns out to be somewhat too low in our EFKM simulations. In this respect, better agreement with experiments might be reached by reducing the density of $c$-electrons further ($\langle \hat{n}_j^c \rangle < 0.1$). In any case, the "flat-band" region falls entirely into the restricted momentum window monitored within the ARPES measurements, see dashed lines in Fig. 2(d). It should be noted at this point that simple BCS-like mean-field approaches will cover the gap opening effect and basically reproduce the dispersions of the coherent part of the spectral functions but fail in giving the correct distribution of the spectral weight and describing the incoherent contributions to the spectra.

If the attraction between electrons and holes is strong, tightly bound excitons will be formed. At the same time the Hartree shift enlarges the orbital splitting and drives the system further into a semiconducting situation [36]. As a result the EI (low-temperature) phase appears as BEC of preformed (excitonic) pairs [6]. This needs to be reflected in the spectral properties of the EFKM as well, which can not longer be described adequately by weak-coupling approaches, such as Hartree-Fock [44], random phase approximation [41] or (projector-based) renormalization methods [25].

Figure 3 shows the different single-particle spectra $A_{c,f}(k,\omega)$, $A(k,\omega)$ and $A^-(k,\omega)$ for the strong-coupling model parameters belonging to the diamond symbol ($\diamond$) in Fig. 1(a). Quite clearly, now the essential characteristics of the spectra are a huge gap for single-particle excitations [see panel (c)], reflecting the fact that tightly bound electron-hole pairs have to be broken, and a strongly flattened top (bottom) of the valence (conduction) band, pointing towards strong band renormalization and a BEC mechanism. Thereby panel (a) [(b)] indicates that the spectral weight rests mainly above [below] the Fermi energy in the inverse-photoemission [photoemission] part of the $c$ [$f$] electrons and is concentrated near the band's center and edges. In view of the pronounced band flattening, the obvious question is whether $Ta_2NiSe_5$ fits into such a EI BEC-type scenario, especially because this material is in truth semiconducting. The answer, however, is no, as can be seen from the results for $A(k,\omega)$ alone. In the

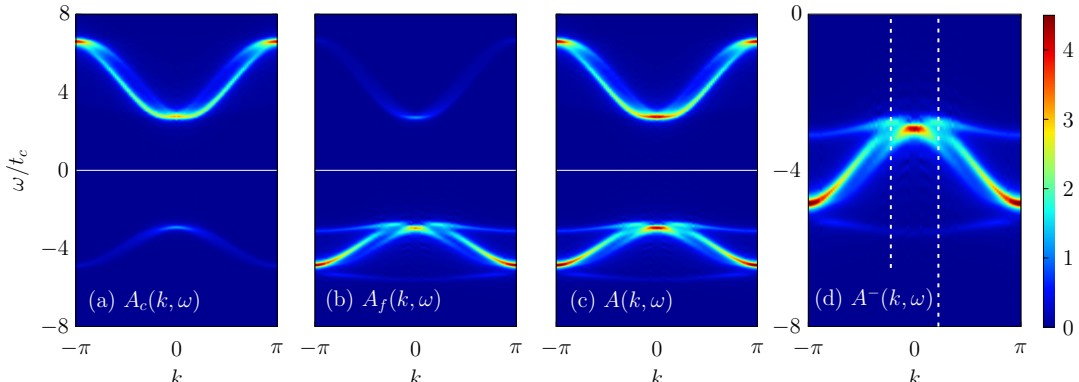

Figure 3: Single-particle spectral functions in the 1D half-filled EFKM with $t_f/t_c = -0.5$, $D/t_c = 0.53$, and $U/t_c = 8$ (simulated with $L_W = 100$).

BEC-EI regime of the EFKM, a two-branch structure emerges in the PES $A^-(k,\omega)$ in the vicinity of the Brillouin zone center ($\Gamma$ point), just as in the strong-coupling regime of Hubbard model (see Fig. 8(d) in Appendix B.2), which is not observed in ARPES. Furthermore, the flat-band region in momentum space appearing in the EFKM PES spectrum is notably wider than that for Ta$_2$NiSe$_5$; cf., the zoomed-in panel Fig. 3(d).

To draw an interim balance, it seems that the ARPES experiments on Ta$_2$NiSe$_5$ can be explained in the framework of the pure 1D EFKM with parameters deep in the weak-coupling region and very close to the EI-BI transition, suggesting an EI state of BCS type.

### 3.3 Photoemission spectra at finite temperatures

To further substantiate that the possible EI state of Ta$_2$NiSe$_5$ might be of BCS type, we now analyze the single-particle spectral functions at finite temperatures. It is of specific interest to prove the leakage of spectral weight of the lower band to energies above the Fermi energy in the EFKM, similar to the melting of the Mott gap in the Hubbard model, see Ref. [53] and Appendix B.2. Such leakage reflects the intensity tail above $E_F$ after pumping observed in a recent t-ARPES experiment for Ta$_2$NiSe$_5$ [19]. Likewise we should verify in terms of the EFKM that the upper band in the PES at $T > 0$ is missing or merged into a single band due to the small value of $U$, just as indicated by ARPES on Ta$_2$NiSe$_5$ [11, 12, 21].

To achieve this technically, we perform finite-temperature simulations, working with a grand canonical ensemble and employing t-DMRG with IBC and an $N_{uc} = 2$ (2 physical and 2 auxiliary sites) unit cell for the iMPS. The chemical potential $\mu$ is fixed by the infinite TEBD so as to fulfil the condition $\langle \hat{n}_j^c \rangle + \langle \hat{n}_j^f \rangle = 1$ for each target temperature, before starting simulations for the dynamical correlation functions.

Figure 4 provides the PES $A^-(k,\omega)$ spectrum of the 1D EFKM for the same parameters as in Fig. 2 but at finite temperatures. Clearly, at low temperatures ($T/t_c = 0.1$ [Fig. 4(a)]) the results differ marginally from their zero-temperature counterparts given in Fig. 2; only the peak height is slightly reduced. Because of the small charge gap, the gap melting occurs already at very low temperatures. This is particularly apparent in the $c$-orbital contribution, which exhibits a significant spectral weight for $|k| \lesssim k_F$ below the Fermi energy at $T = 0$ [see Fig. 2(a)]. If $T$ gets larger than $t_c$, the $c$-orbital band leaks into the region above Fermi energy [Fig. 4(c)] and, as a consequence, $c$ and $f$ bands nearly follow the non-interacting band dispersions $\epsilon_\alpha(k) = -2t_\alpha \cos k$ with $\alpha \in \{c, f\}$, see Fig. 4(d). While ARPES on Ta$_2$NiSe$_5$ has not indicated such a behaviour so far, t-ARPES experiments detect the almost bare-band structure showing spectral weight even above Fermi energy [19]. This further supports that the EI state observed in Ta$_2$NiSe$_5$ is of BCS type. Note that the flat valence band shifts to higher

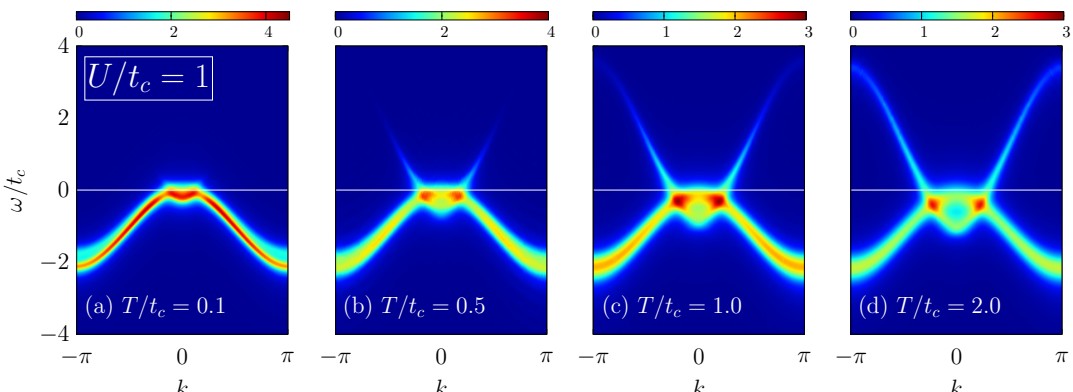

Figure 4: Photoemission spectra $A^-(k, \omega)$ in the 1D half-filled EFKM at finite temperatures. Model parameters are $t_f/t_c = -0.5$, $D/t_c = 2.11$, and $U/t_c = 1$. We use $L_W = 256$.

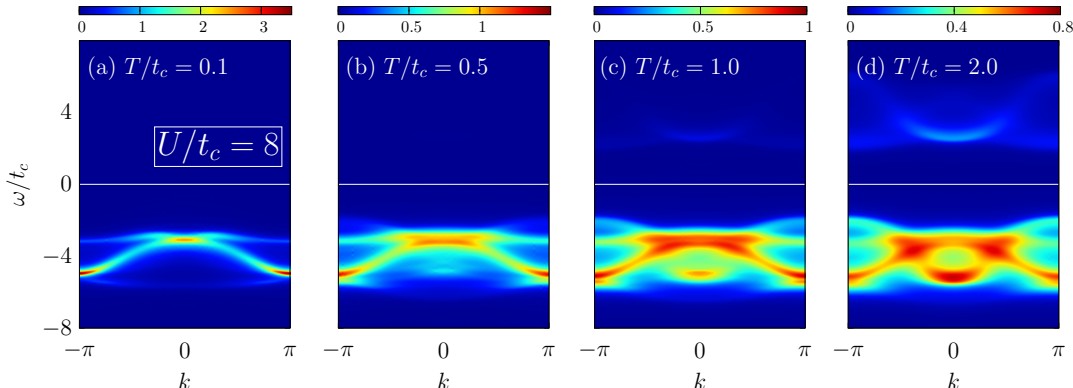

Figure 5: Finite-temperature PES $A^-(k, \omega)$ in the strong-coupling regime of the 1D EFKM with $t_f/t_c = -0.5$, $D/t_c = 0.53$, and $U/t_c = 8$. In the simulations, we use $L_W = 128$.

energies (but is still located below the Fermi energy) when the temperatures is raised [13]. Since we observe no difference between the location of the flat band near $k = 0$ at $T = 0$ and $T/t_c = 0.1$, see Fig. 2(d) and Fig. 4(a), respectively, this might be attributed to the higher dimensionality of the real material.

Let us finally comment on the behavior of the finite-temperature PES in the strong-coupling regime ($U/t_c = 8$; see Fig. 5). Here, by and large, the situation is reminiscent to that in the Hubbard model for $U/t = 8$, cf. Appendix B.2. While, at the low temperature $T/t_c = 0.1$, the PES barely changes compared to $T = 0$, at about $T/t_c = 0.5$, a noticeable part of the spectral weight is redistributed to (large) negative energies [see Fig. 5(b)]. As the temperature is further raised up to $T \simeq J_{\text{eff}} = 4|t_f|t_c/U$, a PES band appears above the Fermi energy [Fig. 5(c)]. It becomes more distinct for $T > J_{\text{eff}}$ [Fig. 5(d)]. Note that in ARPES experiments on $Ta_2NiSe_5$ none of these effects have been reported.

## 4 Conclusions

To sum up, we examined the ground-state and spectral properties of the half-filled extended Falicov-Kimball model (EFKM) in one spatial dimension where mean-field-like approaches usu-

ally fail. Combining the time-dependent density-matrix renormalization group technique with the purification method and infinite boundary conditions, we are able to obtain unbiased numerical results that hold in the thermodynamic limit, i.e., for the infinite system, at both zero and finite temperature.

Regarding the ground-state phase diagram we confirmed the existence of an excitonic insulator state exhibiting quasi-long-range order, sandwiched between staggered-orbital-order and band-insulator phases in the Coulomb-interaction orbital-splitting plane. The excitonic insulator typifies a BCS (Bose-Einstein) macroscopically coherent quantum state for weakly (strongly) coupled $c$ and $f$ electrons. Our analysis of the pair condensation amplitude allows to quantify the BCS-BEC crossover region, where the character of the constituting electron-hole pairs changes from "Cooper-like" (on the semimetal side) to tightly-bound excitons (on the semiconductor side).

The spectral properties of the EFKM are further evidence for a BCS-BEC crossover of the excitonic condensate. Hallmarks of the BCS-BEC crossover are an increasing band renormalization, including a widening of the charge gap, as well as spectral weight transfer (also from the coherent to the incoherent part of the single-particle spectrum) and a weakening of Fermi surface effects.

Most interesting in view of recent ARPES experiments on $Ta_2NiSe_5$ seems to be the band flattening, which has been taken as strong indication for the formation of an excitonic insulator state. Simulating the zero-temperature photoemission in the weak-coupling regime of half-filled EFKM with a parameter set that (independently) has been shown being most suitable for $Ta_2NiSe_5$, the flat valence band is confirmed in the narrow region of momentum space detected by ARPES. Therefore we believe that $Ta_2NiSe_5$ typifies a BCS-type excitonic insulator. Findings that $Ta_2NiSe_5$ behaves semiconductor-like in some circumstances might be because it is located close to the band insulator boundary in the EFKM model parameter space.

At finite (high) temperatures, a signature of the conduction band appears in the photoemission spectra above the Fermi energy. Here, both valence and conduction bands follow a rescaled cosine dispersion with different hopping amplitudes. It would be interesting to estimate the effective temperature after pulse irradiation by comparing our numerical data with experimental results. Pulse irradiation can be treated within the t-DMRG scheme as demonstrated in Ref. [54], i.e., we can carry out a real-time evolution to simulate $A(k,\omega)$ in this case. Such a direct comparison with time-dependent ARPES experiments on $Ta_2NiSe_5$ is left for further studies.

Finally we wish to stress that the investigated EFKM has to be considered as a generic but very simplistic model for the excitonic transition in solids. For example, the interplay between the electronic and lattice instability [49], which is intensively discussed for $Ta_2NiSe_5$ too [55–58], cannot be addressed. Moreover, recent and perhaps more realistic models of $Ta_2NiSe_5$ [59] include also an off-site hybridization: In such a system the excitonic insulator transition arises from the spontaneous breaking of a residual discrete symmetry, rather than by that of a continuous symmetry related to the conservation of the relative charge between valence and conduction electrons [26].

## Acknowledgements

We thank Y. Ohta for fruitful discussions. Density-matrix renormalization-group calculations were performed using the ITensor library [60]. S.E. and F.L. are supported by Deutsche Forschungsgemeinschaft through project EJ 7/2-1 and FE 398/8-1, respectively.

# A  Numerical approach

In this Appendix, we explain how to simulate the time-dependent correlation functions $A^{\pm}(x, t)$ of Eq. (5) by combining the t-DMRG technique with the purification method for matrix-product states (MPS).

The density matrix $\rho(\beta)$ of the system can be regarded as the reduced density matrix of a pure state $|\psi(\beta)\rangle$ in an enlarged Hilbert space, $\rho(\beta) = \mathrm{Tr}_Q|\psi(\beta)\rangle\langle\psi(\beta)|$, where the trace is taken over the space $Q$ spanned by the auxiliary sites. The expectation value of an operator $\hat{O}$ therefore becomes $\langle\hat{O}\rangle_\beta = \langle\psi(\beta)|\hat{O}|\psi(\beta)\rangle$. To determine the equilibrium density matrix at target temperature $T$, we first construct a state $|\psi(0)\rangle$ corresponding to $\rho(\beta = 0)$ and then perform an imaginary time evolution $|\psi(\beta)\rangle = e^{-\beta\hat{H}/2}|\psi(0)\rangle$ on the physical subsystem. For $|\psi(0)\rangle$ in the grand-canonical ensemble, we choose a state with a simple MPS representation, in which each physical site is in a maximally entangled state with an auxiliary site. The time evolution is then carried out, e.g., by using the time-evolving block decimation technique [61, 62] and swap gates [63].

To obtain a high momentum resolution for the spectral functions, the correlation functions need to be calculated up to long enough distances. It is therefore important to consider large systems or even work directly in the thermodynamic limit by using infinite MPS (iMPS) with a translationally invariant unit cell. Following the latter approach, we calculate the purification $|\psi(\beta)\rangle$ of the equilibrium density matrix using the infinite time-evolving block decimation technique [61, 62] with reorthogonalization [64]. Similar to the zero-temperature case, dynamic properties are calculated by switching to real-time evolution after a local perturbation is applied to $|\psi(\beta)\rangle$. The perturbation lifts the translation symmetry of the state so that it can no longer be described by an iMPS with a repeated unit cell. It is sufficient, however, to update only the tensors of the MPS in the finite region over which the perturbation spreads during the time evolution. The resulting algorithm is essentially that for a finite-system with specific infinite boundary conditions (IBC) [52].

To extend the simulated time and thereby increase the resolution in energy space, we furthermore exploit the time-translation invariance as described in Refs. [65, 66]. The idea is to consider two states $\hat{O}_j|\psi(\beta)\rangle$ and $\hat{O}_{j+x}|\psi(\beta)\rangle$, and evolve one forward and one backward in time so that their scalar product yields the correlation function for distance $x$. When doing this, it is important that in addition to the time evolution of the physical sites, the auxiliary sites are evolved in the reverse direction, which also slows down the buildup of entanglement in the purification state [67]. An advantage of IBC, in addition to the absence of finite-size effects, is that we can exploit the spatial translation symmetry of the equilibrium state when calculating the correlation functions with the above method. As demonstrated in Ref. [68], the correlation function at an arbitrary distance can be obtained from a single state just by shifting the two states with forward and backward time evolution relative to each other, while one would need a separate simulation for each distance with open boundary conditions. Special care should be taken in implementing this procedure if the operator $\hat{O}_j$ in Eq. (5) is fermionic as in the calculation of the (I)PES. The Jordan–Wigner strings $\hat{F}_j = \exp(i\pi\hat{n}_j)$ with $\hat{n}_j = \hat{n}_{j,\uparrow} + \hat{n}_{j,\downarrow}$, which appear in the mapping of spinful fermionic operators into spinful bosonic operators, need to be taken into account when preparing the MPS for the real time evolution. We define

$$\hat{c}_{j,\uparrow} = \left(\prod_{\ell=1}^{j-1}\hat{F}_\ell\right)\hat{a}_{j,\uparrow}, \quad \hat{c}_{j,\downarrow} = \left(\prod_{\ell=1}^{j-1}\hat{F}_\ell\right)(\hat{F}_j\hat{a}_{j,\downarrow}), \tag{7}$$

where $\hat{a}_{j,\sigma}^{(\dagger)}$ and $\hat{a}_{j',\sigma'}^{(\dagger)}$ obey fermionic anticommutation relations if $j = j'$ and $\sigma = \sigma'$, and bosonic commutation relations otherwise. For $\hat{O}_j = \hat{c}_{j,\uparrow}$ ($\hat{O}_j = \hat{c}_{j,\downarrow}$), we have to apply the

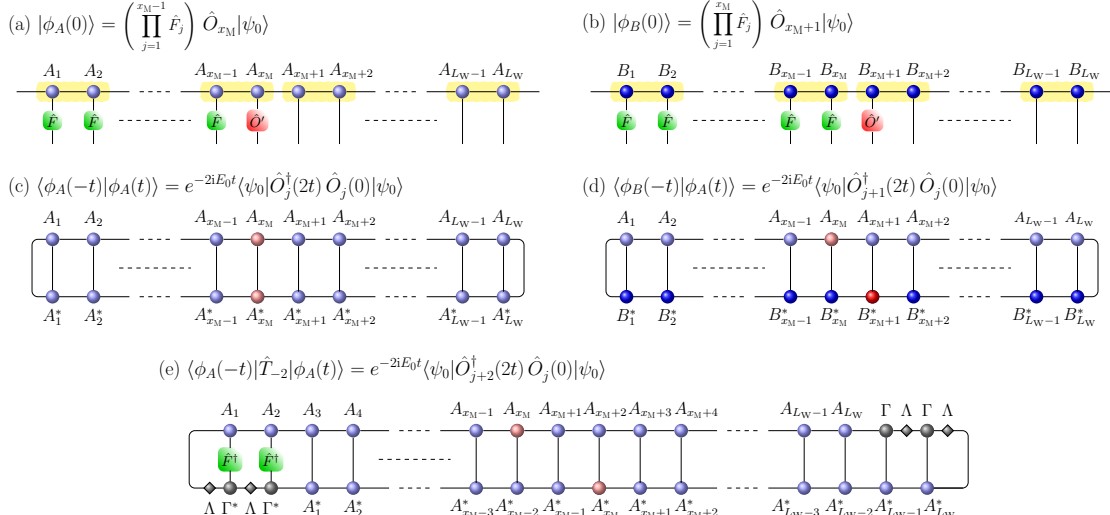

Figure 6: Graphical representation of the iMPS state for the calculation of dynamical correlation functions. The blue circles represent the tensors in the finite window, the squares correspond to the operators to be applied to the states, and the gray tensors in panel (e) represent the iMPS unit cell. Panel (e) shows how the correlation functions can be calculated for arbitrary distance by shifting the states relative to each other, i.e., by inserting an $n$-site translation operator $\hat{T}_n$. For further explanation see text.

mapped operator $\left(\prod_{\ell=1}^{j-1} \hat{F}_\ell\right)\hat{O}_j'$, where $\hat{O}_j' = \hat{a}_{j,\uparrow}$ ($\hat{O}_j' = \hat{F}_j \hat{a}_{j,\downarrow}$). Note, that the Jordan-Wigner strings only act on the physical sites.

In the following, we give the step by step explanation of the complete algorithm (for simplicity at $T = 0$):

0) Simulate the ground state $|\psi_0\rangle$ in iMPS representation using iDMRG [69]. Here, we consider a two-site unit cell iMPS in the canonical form with tensors $\Lambda^{[n]}$ and $\Gamma^{[n]}$ ($n \in \{0,1\}$).

1) Construct an MPS with IBC with appropriate window size $L_W$ by repeating the iMPS. The resulting state can be written as

$$|\phi(t=0)\rangle = \sum_{\boldsymbol{\sigma}} \cdots \Lambda^{[1]}\Gamma^{[0]\sigma_0}A^{[1]\sigma_1}\cdots A^{[L_W]\sigma_{L_W}}\Gamma^{[1]\sigma_1}\Lambda^{[1]}\cdots|\boldsymbol{\sigma}\rangle,$$

where $\sigma_j$ denotes the basis states of the local Hilbert space at site $j$.

2) Construct the wave functions $|\phi_A(0)\rangle = \hat{O}_{x_M}|\phi(0)\rangle = \left(\prod_{\ell=1}^{x_M-1} \hat{F}_\ell\right)\hat{O}_{x_M}'|\phi(0)\rangle$ and $|\phi_B(0)\rangle = \hat{O}_{x_M+1}|\phi(0)\rangle = \left(\prod_{\ell=1}^{x_M-1} \hat{F}_\ell\right)\hat{O}_{x_M+1}'|\phi(0)\rangle$ with $x_M \equiv L_W/2$, see Figs. 6(a) and (b), respectively.

3) Evolve both states in time by TEBD as $|\phi_{A/B}(t)\rangle = e^{-i(t/2)\hat{H}}\hat{O}_{x_M/x_M+1}|\phi(0)\rangle$.

4) Evaluate two-point correlators $\langle\hat{O}_{j+r}^\dagger(t)\hat{O}_j(0)\rangle$ by shifting $|\phi_{A/B}\rangle$ relative to each other, see, e.g., Fig. 6(c) for $r = 0$, (d) for $r = 1$ and (e) $r = 2$. Extra attention should be paid to Jordan-Wigner strings $\hat{F}^\dagger$ for the two-site iMPS in panel (e).

After step 4 we go back to step 3 until the target time is reached. At finite temperatures this process can be performed in an analogue manner. Finally, obtained data can be extrapolated to longer times through linear prediction [70]. We apply an exponential windowing function

to the obtained time-dependent correlation functions, which, after the Fourier transform, corresponds to a convolution of the spectral functions by a Lorentzian $\eta_L/(\pi(\omega^2 - \eta_L^2))$ with the broadening parameter $\eta_L$. In Appendix B we will exemplarily calculate the spectral functions in the Hubbard model by means of the presented t-DMRG-based scheme. Again the time step $\tau = \tau_\beta = 0.01$ and the maximum bond dimension is 800 so that the truncation error is less than $10^{-6}$.

# B  Spectral functions in the Hubbard model

## B.1  Model Hamiltonian

The 1D Hubbard model reads

$$\hat{H} = -t_h \sum_j \left( \hat{c}^\dagger_{j,\sigma} \hat{c}_{j+1,\sigma} + \text{H.c.} \right) + U \sum_j \left( \hat{n}_{j,\uparrow} - \tfrac{1}{2} \right) \left( \hat{n}_{j,\downarrow} - \tfrac{1}{2} \right), \tag{8}$$

where $\hat{c}^\dagger_{j,\sigma}$ ($\hat{c}_{j,\sigma}$) creates (annihilates) a fermion with spin projection $\sigma \in \{\uparrow, \downarrow\}$ at lattice site $j$, and $\hat{n}_{j,\sigma} = \hat{c}^\dagger_{j,\sigma} \hat{c}_{j,\sigma}$. Remarkably, at zero temperature, the model (8) can be solved by the Bethe ansatz [71]. Let us again consider the half-filled case where the number of particles $N$ is equal to the number of lattice sites $L$. In this case, the model has a Mott insulating state for any $U > 0^+$ with a finite charge gap that increases exponentially with $U$ in the weak-coupling regime and grows linearly for large interactions. The spin-degree-of-freedom excitations, however, are gapless, so that the spin-spin correlations decay with a power-law.

The momentum-resolved spectrum of the physical excitations can be obtained by considering two different types of elementary excitations: gapped spinless excitations carrying charge $\pm e$ called holons ($h$) and antiholons ($\bar{h}$), respectively, and gapless charge-neutral excitations carrying spin $\pm 1/2$ called spinons ($S^z = 1/2$ spinon $s$ and $S^z = -1/2$ spinon $\bar{s}$). At half filling, the Bethe ansatz yields the following dressed energies and momenta for these excitations in the thermodynamic limit [71]:

$$\mathcal{E}_{s/\bar{s}}(\Lambda) = 2 \int_0^\infty \frac{d\omega}{\omega} \frac{J_1(\omega) \cos(\omega \Lambda)}{\cosh(\omega U/4)}, \tag{9}$$

$$\mathcal{P}_{s/\bar{s}}(\Lambda) = \frac{\pi}{2} - \int_0^\infty \frac{d\omega}{\omega} \frac{J_0(\omega) \sin(\omega \Lambda)}{\cosh(\omega U/4)}, \tag{10}$$

$$\mathcal{E}_{h/\bar{h}}(k) = 2 \cos k + \frac{U}{2} + 2 \int_0^\infty \frac{d\omega}{\omega} \frac{J_1(\omega) \cos(\omega \sin k) e^{-\omega U/4}}{\cosh(\omega U/4)}, \tag{11}$$

$$\mathcal{P}_h(k) = \mathcal{P}_{\bar{h}}(k) + \pi = \frac{\pi}{2} - k - 2 \int_0^\infty \frac{d\omega}{\omega} \frac{J_0(\omega) \sin(\omega \sin k)}{1 + \exp(|\omega| U/2)}, \tag{12}$$

where $J_n(\omega)$ are $n$-th-order Bessel functions. The dispersion relations are obtained by varying the parameters $\Lambda \in (-\infty, \infty)$ and $k \in (-\pi, \pi]$. The physical excitations follow from permitted combinations of the elementary excitations. For example, the spin-charge scattering states whose energy and momentum are given by:

$$E_{SC}(\Lambda, k) = \mathcal{E}_s(\Lambda) + \mathcal{E}_h(k), \quad P_{SC}(\Lambda, k) = \mathcal{P}_s(\Lambda) + \mathcal{P}_h(k). \tag{13}$$

In the following section B.2, our numerical approach will be benchmarked against the Bethe ansatz results (9)-(13).

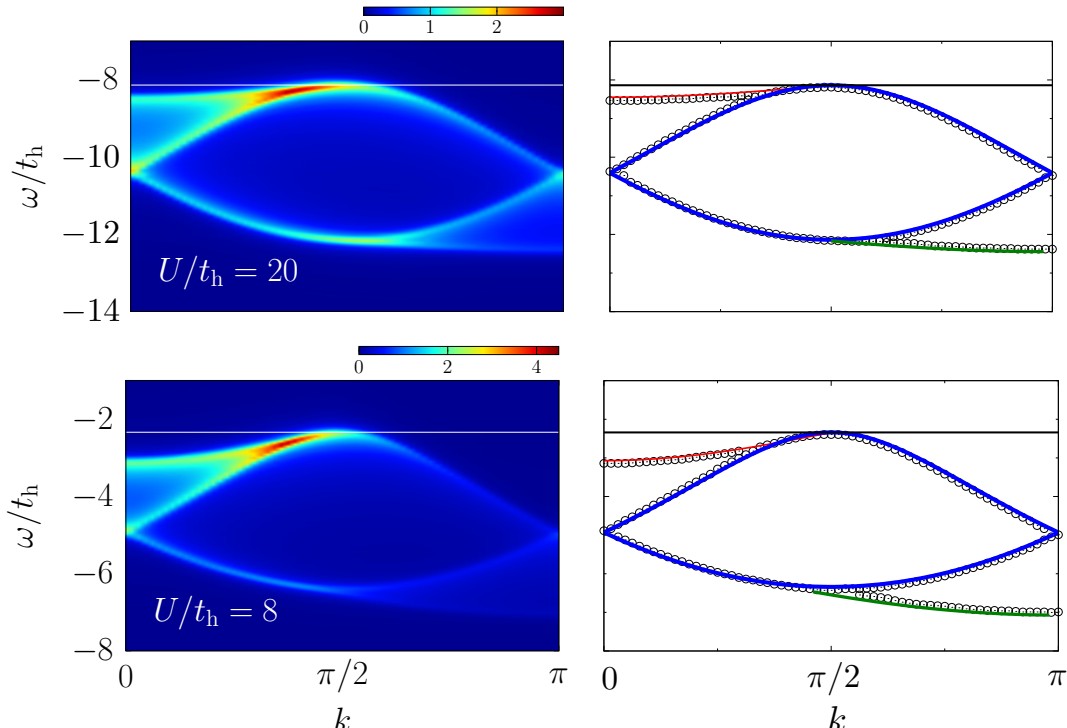

Figure 7: Photoemission spectra in the strong-coupling regime of half-filled 1D Hubbard model with $U/t_{\rm h} = 20$ (top) and 8 (bottom). Results in the left panels and symbols in the right panels are obtained by t-DMRG. The blue (red) line denotes the holon (spinon) branch, the green line gives the lower onset of the spinon-holon excitation continuum according to the Bethe-ansatz solution in the thermodynamic limit (9)-(13). The white and black horizontal lines indicate half of the single-particle gap $-\Delta_{\rm c}/2t_{\rm h}$.

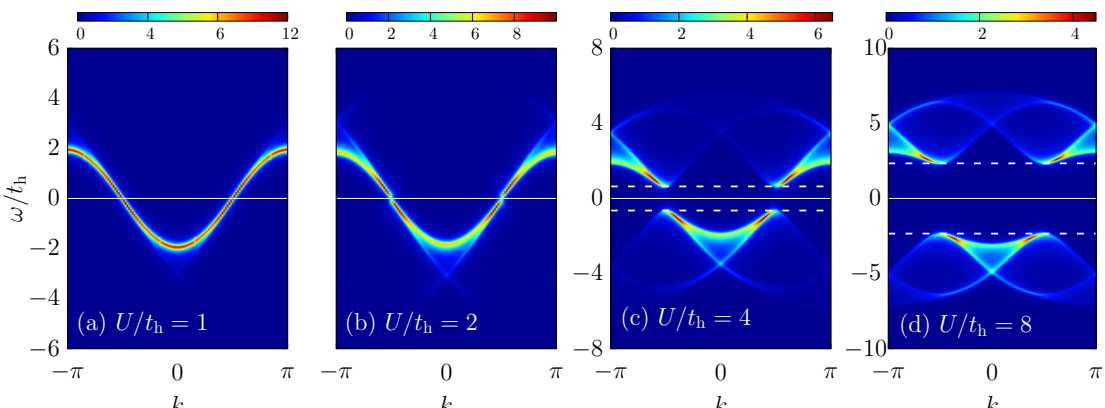

Figure 8: $A(k, \omega)$ in the half-filled 1D Hubbard model for various Coulomb repulsions $U$ at $T = 0$. Dashed lines in (c) and (d) mark the Bethe-ansatz single-particle gap $\pm\Delta_{\rm c}/2$.

## B.2 Spectral functions at zero temperature

Figure 7 presents the PES, $A^-(k, \omega)$, for a half-filled Hubbard chain in the strong-coupling regime with $U/t_{\rm h} = 20$ (upper panels) and 8 (lower panels) at $T = 0$. Data are obtained by the t-DMRG technique with IBC. We observe two separately dispersing peaks in the momentum

interval $[0, \pi/2]$ that merge at the Fermi momentum $k_F = \pi/2$ where the gap opens. In the interval $[\pi/2, \pi]$ we also find two dispersive peaks merging at the zone boundary. Comparing the course of the t-DMRG peak positions with the exact Bethe-ansatz dispersions (9)-(13), we are able to identify the physical nature of each dispersion, see right panels of Fig. 7. Since agreement is excellent, not only in the limit of large Coulomb interaction $U/t_h = 20$ but also for $U/t_h = 8$, we can clearly assign the excitations as spinon (red lines) or holon (blue lines) branches, and can also identify the onset of the holon-spinon continuum (green lines). As an advantage, the t-DMRG data also provide the spectral weight as a function of momentum (cf., the color code). Because the spinon bandwidth is about the effective exchange coupling $J_{eff} = 4t_h^2/U$, it is almost flat for very large $U$ and gets wider when $U$ shrinks (cf. upper and lower panels). We would like to point out that the upper onset of the peaks of $A^-(k, \omega)$ nicely agree with half of the single-particle gap $\Delta_c/2$, see white and black lines.

The whole single-particle spectrum $A(k, \omega)$, which includes also the IPES contribution, is shown Fig. 8 for interaction strengths ranging from $U/t_h = 1$ to 8. Since the Mott gap $\Delta_c$ is exponentially small in the weak-coupling regime $[\Delta_c/t_h \simeq 0.005\ (0.173)$ for $U/t_h = 1$ (2)], the dispersion for $U/t_h = 1$ [Fig. 8(a)] is practically identical to that of the bare cosine band with the holon bandwidth $W_h = 4$, without opening a gap due to the finite Lorentzian broadening ($\eta_L/t_h = 0.1$). Increasing $U$ slightly, both spinon and antiholon branches appear, see Fig. 8(b) for $U/t_h = 2$. In the intermediate-to-strong-coupling regime, also the opening of the charge gap $\Delta_c$ at the Fermi momenta $k = \pm k_F$ becomes visible [cf. Figs. 8(c) and (d)]. The spectral weight of the PES (IPES) is mainly located in the antiholon and spinon branches in the interval $[-k_F, k_F]$ ($[-\pi, -k_F]$ and $[k_F, \pi]$). We note that $A(k, \omega)$ for $U/t_h = 4$ was previously discussed within t-DMRG in Ref. [53], but only for finite systems. Our infinite-system data validate these results to a large extent.

### B.3 Photoemission spectra at finite temperatures

Figure 9 demonstrates the temperature effects on PES in the half-filled 1D Hubbard model at various Coulomb interaction strengths obtained by combining the t-DMRG technique with the purification method as described in Appendix A. Clearly, at low temperature ($T/t_h = 0.1$) the spectral functions are very similar to their zero-temperature counterparts, whereby the peak maxima are slightly reduced (see the different scales of the color bars with increasing temperature). In the strong-coupling regime, the two-brunch structure of the spectral function only survives up to $T \sim J_{eff}$ (cf. the panels for $U/t_h = 4$ and 8 in Fig. 9). Furthermore, the "$\overline{V}$-shaped" structure of the spectrum and the concentration of spectral weight around $\omega = -U/2 - 2t_h$, $k = 0$ and $\omega = -U/2 + 2t_h$, $k = \pm \pi$ found in the strong-coupling approach [53] is corroborated by our numerical results. A signature of upper Hubbard band is also visible here, which can be qualitatively described by the Hubbard-I mean-field approximation [72, 73] (note that we have shown only the PES part of $A(k, \omega)$ in Fig. 9).

Most interesting seems to be the melting of the Mott gap in the vicinity of the Fermi level: The zero-temperature PES spectral weight leaks into the Mott gap regime in the doped Mott insulator (two holes in the finite-size system), which leads to the extension of the spinon-antiholon continuum [53]. Similarly, holes in the lower Hubbard band left behind as a result of doublon creations can be occupied by thermally excited quasiparticles. This leads to the melting of the Mott gap with increasing temperature, which becomes more distinct when decreasing the Coulomb repulsions $U$.

At very high temperatures ($T > t_h$), the energy distance between upper and lower Hubbard bands is insignificant (provided $U$ is not too large) and finally both bands merge into a single band $\epsilon(k) = -2t_h \cos k$, see Fig. 9(p).

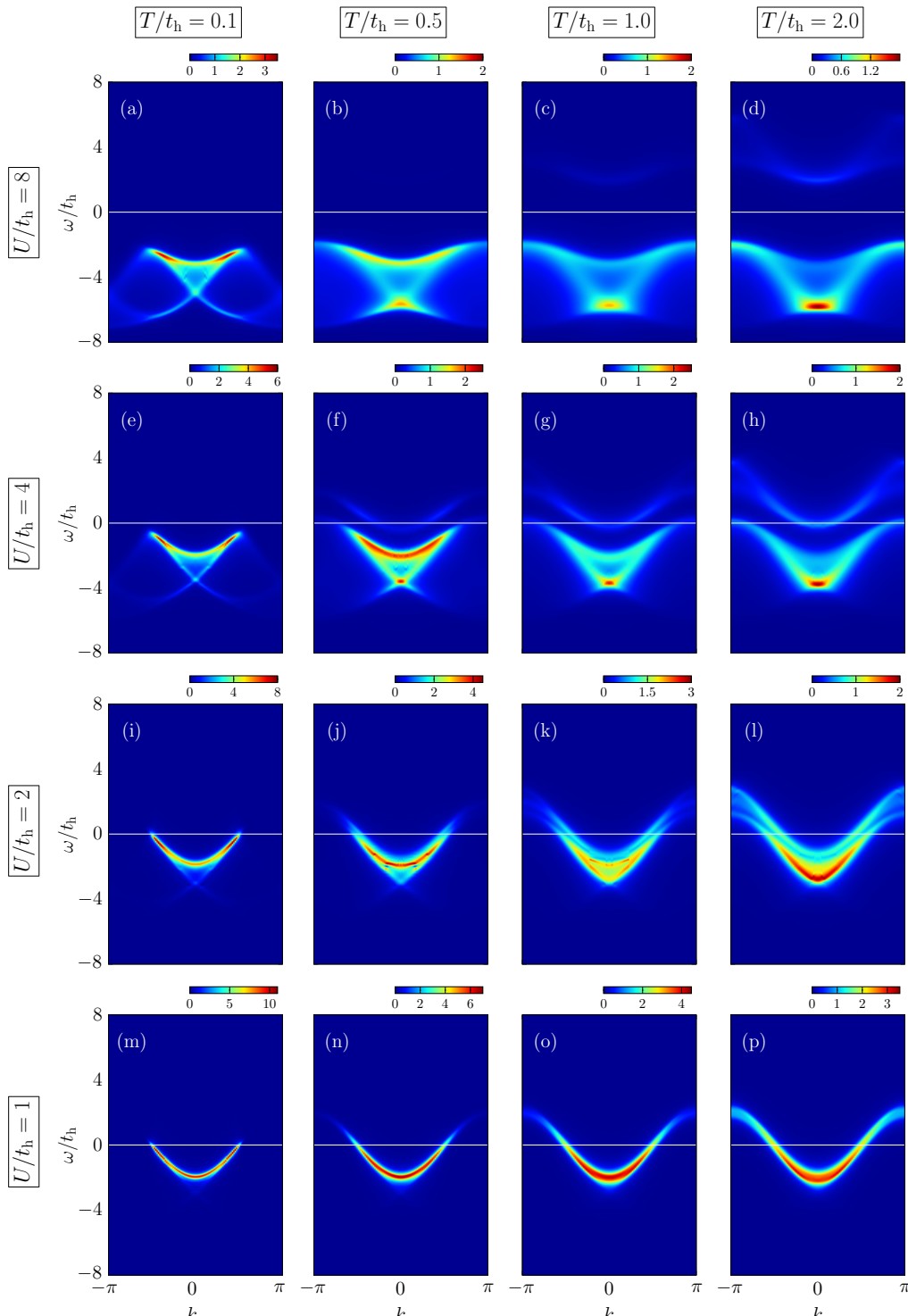

Figure 9: Photoemission spectra $A^-(k,\omega)$ in the half-filled 1D Hubbard model for various Coulomb repulsions $U/t_\mathrm{h}$ at different temperatures $T/t_\mathrm{h}$.

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
