# Peer review of "Finite-temperature photoemission in the extended Falicov-Kimball model: a case study for Ta$_2$NiSe$_5$"

_SciPost Physics, doi:SciPost Phys. 10, 077 (2021)_

## Round 1 · Referee Report · Anonymous (Referee 1) · 2021-2-7

Report

The authors use an unbiassed time-dependent DMRG technique to compute the photo-emission spectra of an extended Falicov-Kimball model at zero and finite temperatures.
The main goal of the work is to characterize the excitonic insulator Ta2NiSe5 that undergoes a continuous orthorhombic to monoclinic transition at T_c ~328 K. Based on their calculations, the authors conclude that Ta2NiSe5 becomes an excitonic insulator of BCS-like type (weak-coupling regime).
The manuscript is well written, and the results are physically sound and relevant for Ta2NiSe5. Although the notion of excitonic insulator has been known for 60 years, the main contribution of this work is important because it is not easy to find experimental realizations of this idea.
In view of these observations, I recommend this manuscript for publication in SciPost after the authors consider the following recommendations:

The model proposed by the authors has a U(1) continuous symmetry associated with the conservation of the difference between total charges in the f and c orbitals. This is the symmetry that is spontaneously broken in the excitonic state. However, this symmetry is not present in real materials. In other words, the exciton current does not obey a continuity equation in real materials. A more realistic model of Ta2NiSe5 should include an off-site hybridization that explicitly breaks the above-mentioned U(1) symmetry. In such a description, the excitonic insulator arises from the spontaneous breaking of a residual discrete symmetry, as it is clear from the orthorhombic to monoclinic transition of Ta2NiSe5. I recommend the authors to clarify this point for readers that are not familiarized with the approximations that are made when the EFKM [Eq.(1) of the manuscript] is adopted as a minimal model of a real material.

In section 3.2, the authors say: “in a strictly 1D model system, a symmetry-broken EI phase can only exist at zero temperature”. The U(1) symmetry that is spontaneously broken in the EI phase is not broken at any temperature (including T=0) for a 1D system. The ground state of model (1) has a non-zero superfluid density that can be associated with the EI, but the U(1) symmetry is not spontaneously broken in the thermodynamic limit: the expectation value of the order parameter is equal to zero even at T=0 (the correlations decay as a power law and we say that the system develops quasi-long range order at T=0).

One of the main conclusions of the manuscript is that Ta2NiSe5 is in the BCS regime. In view of that observation, I wonder if the authors can compare their DRMG results presented in Figs. 2 and 4 against the results that are obtained from a simple BCS mean field theory. Note also that the simple mean-field BCS theory can be applied to more realistic 3D versions of the model, which cannot be treated with the DMRG technique.
  • validity: high
  • significance: good
  • originality: good
  • clarity: high
  • formatting: excellent
  • grammar: excellent

Author:  Satoshi Ejima  on 2021-02-22  [id 1255]

(in reply to Report 1 on 2021-02-07)

We thank the referee for her/his expert report and the valuable comments that have led us to clarify some points.

(i) The model proposed by the authors has a U(1) continuous symmetry associated with the conservation of the difference between total charges in the f and c orbitals. This is the symmetry that is spontaneously broken in the excitonic state. However, this symmetry is not present in real materials. In other words, the exciton current does not obey a continuity equation in real materials. A more realistic model of Ta2NiSe5 should include an off-site hybridization that explicitly breaks the above-mentioned U(1) symmetry. In such a description, the excitonic insulator arises from the spontaneous breaking of a residual discrete symmetry, as it is clear from the orthorhombic to monoclinic transition of Ta2NiSe5. I recommend the authors to clarify this point for readers that are not familiarized with the approximations that are made when the EFKM [Eq.(1) of the manuscript] is adopted as a minimal model of a real material.

Let us first point out that in the EFKM under study the difference between the f- and c-electrons (N_f and N_c, respectively) is not fixed (in the EI phase). We only fixed the total number of electrons (N=N_f+N_c) in the system as N/L=1 (half-filled band case) with L being the number of lattice sites. We furthermore note that in a strictly 1D system the expectation value of the order parameter <c^\dagger f> is always zero in the absence of an explicit c-f-band hybridization, which is also related to the next question by the Referee. We clarified this point in the paragraph above Eq. (4) and at the end of the first paragraph of Sec. 3.2. Moreover, in the new last paragraph of the conclusions (Sec. 4), we comment on a perhaps more realistic modeling of Ta$_2$NiSe$_5$ (taking into account e.g. an off-site hybridization), mentioned the possibility of off-site hybridization breaking a discrete symmetry (rather than a global U(1) symmetry), and related this to our model/approximations.
In this context, we added new references [54-58].

(ii) In section 3.2, the authors say: “in a strictly 1D model system, a symmetry-broken EI phase can only exist at zero temperature”. The U(1) symmetry that is spontaneously broken in the EI phase is not broken at any temperature (including T=0) for a 1D system. The ground state of the model (1) has a non-zero superfluid density that can be associated with the EI, but the U(1) symmetry is not spontaneously broken in the thermodynamic limit: the expectation value of the order parameter is equal to zero even at T=0 (the correlations decay as a power law and we say that the system develops quasi-long range order at T=0).

We thank the Referee for pointing out this wrong formulation. We removed the word “symmetry-broken” from this sentence in Sec. 3.2. We discussed the situation in the EFKM in more detail between Eqs. (2) and (3), with regard to the vanishing <c^\dagger f> expectation value (in the absence of an explicit c-f-band hybridization) and concerning the characteristic power-law excitonic correlations in a strictly 1D system.

(iii) One of the main conclusions of the manuscript is that Ta2NiSe5 is in the BCS regime. In view of that observation, I wonder if the authors can compare their DRMG results presented in Figs. 2 and 4 against the results that are obtained from a simple BCS mean-field theory. Note also that the simple mean-field BCS theory can be applied to more realistic 3D versions of the model, which cannot be treated with the DMRG technique.

We have pointed out at the end of page 7 that in the weak-coupling regime, simple BCS-like approaches will cover the gap opening effect and basically reproduce also the dispersions of the coherent part of the spectral functions but fail in giving the correct distribution of the spectral weight and describing the incoherent contributions to the spectra. Clearly the strong-coupling regime is not accessible by such mean-field approaches. We also note that mean-field [43,44] and slave-boson [34,35] approaches cannot be used to determine the EI phase in the 1D EFKM system, mainly because there is no continuous symmetry that is broken. Therefore it was the primary concern of this work to provide unbiased numerical results for the ground-state and spectral properties if the strictly 1D EFKM in the whole parameter regime, also in the thermodynamic limit and at both zero and finite temperatures. Hence, by means of our approximation-free treatment, we can figure out which of the experimental results can be explained in the framework of the 1D EFKM physics, and which cannot. Furthermore, we clarified the shortcomings of the BCS approach throughout the text, e.g., on page 7 last paragraph, page 8 end of first paragraph and on page 10 (first paragraph of the conclusions).

---

## Round 1 · Referee Report · Anonymous (Referee 2) · 2021-2-15

Report

The authors present results for the dynamical spectral functions of the extended 1D Falicov-Kimball model at zero and finite temperatures, which serve to model photoemission spectra of quasi-1D materials such as Ta2NiSe5, for which it realizes a minimal model. The results are obtained using matrix product state (MPS) approaches, which are best suited for situations such as the ones studied in the present manuscript. The research focuses on the question of the formation of excitonic insulators in the BEC or BCS regime, respectively, and makes predictions, which will be helpful for the interpretation of experiments on such materials. The paper itself is well written and discusses various interesting results, which should be published. However, in its present form, the authors need to consider the following points prior to publication:

  • it is unclear to me, to which extend a finite-size analysis might be relevant in particular for Fig. 1a. The results shown are obtained for L=60 sites. Could, e.g., the crossover region become a sharp transition in the thermodynamic limit?

  • when discussing BEC-type behavior, I wonder about the results displayed, e.g., in Fig. 3: as I understand, the results are obtained for an infinite system (with IBC/iDMRG), for which the occupation of the corresponding k-mode in the presence of a (quasi-)condensate should diverge. However, the authors report a peak height ~1. Is there a simple explanation?

  • several review articles have not been mentioned, even though they discuss the computation of spectral functions and finite-temperature properties using MPS-type approaches (albeit for finite systems). For example, one misses reference to S. Paeckel et al., Annals of Physics 411, 167998 (2019), which discusses both, the computation of spectral functions, as well as finite-T properties using time-dependent MPS , or Annals of Physics 326, 96 (2011), and possibly further reviews on the computation of dynamical properties using MPS (e.g., Prog. Theor. Phys. Suppl. 176 (2008) ). This is also of relevance for the discussions in Sec. 3, where a broadening is apparently used, but not introduced in Eq. (6) (compare to the discussion in the aforementioned review by Paeckel et al.). It is discussed in the appendices, but the origin of this broadening is unclear, unless one explicitly searches for it in the appendices - the reader would benefit from a clearer presentation of this point.

  • In Fig. 1, one should explain the abbreviations in the figure caption.

  • On page 6 "...distinguish the FEKM from the simpler Hubbard-model case...": this statement is true at half filling, it would be good to mention this.

  • validity: -
  • significance: -
  • originality: -
  • clarity: -
  • formatting: -
  • grammar: -

Author:  Satoshi Ejima  on 2021-02-22  [id 1256]

(in reply to Report 2 on 2021-02-15)

We thank the referee for her/his expert report and the valuable comments that have led us to clarify some points.

(i) it is unclear to me, to which extend a finite-size analysis might be relevant in particular for Fig. 1a. The results shown are obtained for L=60 sites. Could, e.g., the crossover region become a sharp transition in the thermodynamic limit?

As demonstrated in Ref.[36], the BI-EI and SOO-EI phase transition lines can be determined by DMRG by extrapolating the results with finite systems to the thermodynamic limit. By contrast the shaded BCS-BEC crossover region in Fig. 1 was deduced from the condensation amplitude F(k) calculated for a fixed system size L=60 and periodic boundary conditions. However, we checked that the finite-size effects are negligible, e.g., if we compare the displayed results with those for L=40 the change of the shaded region is almost invisible within the scale of Fig.1. We stressed this point in the caption of Fig.1. We also would like to point out that the BCS-BEC transition is a crossover (and no sharp phase transition), even in the thermodynamic limit ($L\to \infty$).

(ii) when discussing BEC-type behavior, I wonder about the results displayed, e.g., in Fig. 3: as I understand, the results are obtained for an infinite system (with IBC/iDMRG), for which the occupation of the corresponding k-mode in the presence of a (quasi-)condensate should diverge. However, the authors report a peak height ~1. Is there a simple explanation?

In Fig. 3, the single-electron spectral functions are shown. To see the divergence of the exitonic momentum distribution function, one should consider the spectral function for the exciton operators b_j^\dagger=c_j^\dagger f_j.

(iii) several review articles have not been mentioned, even though they discuss the computation of spectral functions and finite-temperature properties using MPS-type approaches (albeit for finite systems). For example, one misses reference to S. Paeckel et al., Annals of Physics 411, 167998 (2019), which discusses both, the computation of spectral functions, as well as finite-T properties using time-dependent MPS , or Annals of Physics 326, 96 (2011), and possibly further reviews on the computation of dynamical properties using MPS (e.g., Prog. Theor. Phys. Suppl. 176 (2008) ). This is also of relevance for the discussions in Sec. 3, where a broadening is apparently used, but not introduced in Eq. (6) (compare to the discussion in the aforementioned review by Paeckel et al.). It is discussed in the appendices, but the origin of this broadening is unclear, unless one explicitly searches for it in the appendices - the reader would benefit from a clearer presentation of this point.

In the revised manuscript we give credit to the above review articles, see new Refs [28-32]. Moreover, in order to obtain the spectral functions by the time-dependent DMRG a damping factor e^{-\eta_{L}|t|} is needed due to the finite time range of the simulations. This leads to a Lorentzian broadening in the frequency space. We have corrected Eq. (6) accordingly and added some explanations for this below Eq.(6).

(iv) In Fig. 1, one should explain the abbreviations in the figure caption.

We give the meaning of the abbreviations in the caption of Fig.1 once more.

(v) On page 6 "...distinguish the FEKM from the simpler Hubbard-model case...": this statement is true at half filling, it would be good to mention this.

Thanks for pointing this out, we have specified the sentence above Eq. (5) accordingly.

Anonymous on 2021-03-13  [id 1306]

(in reply to Satoshi Ejima on 2021-02-22 [id 1256])

The corrections and replies address my issues and the manuscript can be published. The only minor point, which might need clarification is concerning the new Ref. [28] "Program of Yukawa International Seminar (YKIS) 2007: Interaction and Nanostructural Effects in Low-Dimensional Systems, Prog. Theor. Phys. Supp. 176 (2008)", which is addressed as "further developments" of the DMRG method on page 3. I cannot see this when looking at the reference (at least it's not obvious, and not a standard reference to cite when thinking of developments of the DMRG), the authors should double check.

---

## Round 2 · Referee Report · Anonymous (Referee 1) · 2021-2-22

Report

The authors have properly addressed the concerns that I listed in my first report. Consequently, I recommend the new version of the manuscript for publication in Sci Post.

---

## Editorial Decision

published